# A Systematic Review of the Key Predictors of Progression and Mortality of Rheumatoid Arthritis-Associated Interstitial Lung Disease

**DOI:** 10.3390/diagnostics14171890

**Published:** 2024-08-28

**Authors:** Laura Groseanu, Cristina Niță

**Affiliations:** 1Department of Rheumatology, Carol Davila University of Medicine and Pharmacy, 020021 Bucharest, Romania; maria.groseanu@umfcd.ro; 2Sfanta Maria Clinical Hospital, 010024 Bucharest, Romania

**Keywords:** rheumatoid arthritis, interstitial lung disease, progression

## Abstract

Background: Rheumatoid arthritis-associated interstitial lung disease (RA-ILD) is an important extra-articular manifestation of rheumatoid arthritis (RA). Identifying patients at risk of progression and death is crucial for improving RA-ILD management and outcomes. This paper explores current evidence on prognostic factors in RA-ILD. Methods: We conducted a systematic literature review to examine the impact of clinical, radiological, and histological factors on lung function decline and the survival of RA-ILD patients. We searched electronic databases, including Medline and EMBASE, from inception to date. The incidence and prognosis of predictors were qualitatively analyzed, and univariate results were combined when feasible. Following the “Preferred Reporting Items for Systematic Reviews and Meta-Analyses (PRISMA)” guidelines, our systematic literature review involved a five-step algorithm. Out of 2217 records, 48 studies were eligible. These studies reported various prognostic factors, including demographic variables, clinical risk factors, serum markers, and preexisting treatments. Results: Lung function declined over time in 1225 subjects, with significant variability in smoking history and radiological/pathological UIP patterns. Severe lung fibrosis and abnormal pulmonary function tests (PFTs) were key univariate prognostic indicators, while age at initial presentation, RA disease activity, predicted DLCO percentage, and UIP pattern were the most reliable multivariate risk factors for ILD progression. Age, male gender, disease duration, RA activity, acute phase reactants, and specific serum biomarkers (Krebs vin den Lungen 6, surfactant protein D, and interleukin 6) were significantly associated with all-cause mortality. Conclusions: RA-ILD is a severe complication of RA characterized by significant prognostic variability. Key prognostic factors include extensive fibrosis observed on imaging, a marked decline in lung function, high RA disease activity, and specific biomarkers. These factors can guide treatment strategies and improve patient outcomes.

## 1. Introduction

Rheumatoid arthritis (RA) affects up to 1% of the global population, primarily women, with a female/male ratio of 3:1 to 4:1 [1]. Rheumatoid arthritis-associated interstitial lung disease (RA-ILD) is a severe and potentially life-threatening manifestation of rheumatoid arthritis (RA), ranking as the second leading cause of death in RA patients after cardiovascular disease [2,3,4,5]. The prevalence of RA-ILD among RA patients ranges from 3.6 to 58%, reflecting variability due to differences in detection methods and thresholds used in various studies [6,7]. The clinical course of RA-ILD is highly variable and generally associated with a worse prognosis compared to RA without lung involvement [8,9]. Notably, patients with the usual interstitial pneumonia (UIP) subtype of RA-ILD have survival rates comparable to those with idiopathic pulmonary fibrosis (IPF) [3].

Recent studies highlight the need for more research to confidently characterize the prognosis and mortality associated with RA-ILD. For instance, a comprehensive review by the European Respiratory Society discussed the epidemiology, risk factors, and diagnostic approaches for RA-ILD, emphasizing the complexity of managing this condition [3]. Additionally, Yang et al. explored new trends and potential hotspots in the research of RA-ILD and provided an in-depth understanding of the development of RA-ILD publications throughout the previous two decades [10]. Despite the growing body of literature, the natural history of RA-ILD remains insufficiently characterized, necessitating further research to better understand prognosis and mortality.

Identifying patients at risk of near-term progression and death is crucial for informed treatment decisions [5,6]. The predictors of disease progression and survival include physiological, radiological, and histopathological features, along with demographic factors and RA severity indicators such as baseline pain, disease activity, and health-assessment scores [3,9,11,12,13,14,15,16,17,18,19,20,21]. Although multifaceted scoring systems exist to estimate prognosis, they are often cumbersome and focused mainly on idiopathic pulmonary fibrosis (IPF), limiting their clinical use [15,22,23]. Relying on the most reliable prognostic indicators—abnormal pulmonary function tests (PFTs), fibrotic extent on imaging, and symptom worsening—often delay treatment initiation until overt fibrotic progression occurs [24]. Additionally, recent evidence suggest that ultrasound, a low-cost point-of-care tool with a high negative predictive value, is becoming valuable for monitoring RA patients [25]. It may also help discriminate ILD severity between males and females, alongside rheumatoid factors, though these findings are still preliminary [26]. Therefore, to tailor treatment with a personalized approach, we need to understand which groups of RA-ILD patients will benefit most from immunosuppression, antifibrotics, or a combination of both as the initial therapeutic strategy [27]. This paper reviews clinical, radiological, and histological data on RA-ILD prognosis and mortality to enhance treatment predictions and optimize care strategies.

## 2. Materials and Methods

This review was conducted and reported according to the Preferred Reporting Items for Systematic Reviews and Meta-Analyses (PRISMA: we did not register for this research) [28]. The objective of the systematic literature review (SLR) was to evaluate the current evidence on the risk factors linked to disease progression and mortality in RA-ILD patients, focusing on previously identified poor outcomes. Specifically, the review aimed to consider any clinical information—such as demographic features, symptoms, PFTs, radiological findings, laboratory tests, and prior treatments—that could influence these outcomes.

### 2.1. Eligibility

The included studies were randomized controlled trials, cohort studies, case/control studies of adults with RA-ILD, and conference abstracts with relevant outcomes. Exclusion were editorial letters, studies without a clear definition of RA-ILD, non-English language studies, and duplicates.

The participants were adults over 18, meeting either the 1987 American College of Rheumatology classification criteria [29] or 2010 American College of Rheumatology/European League Against Rheumatism classification criteria for RA [30] with no restrictions on gender, ethnicity, or disease duration/severity. ILD was characterized by interstitial inflammatory and fibrotic changes in pulmonary parenchyma, diagnosed through symptomatic, functional, radiological, and/or pathological findings [31]. We focused on the most common high-resolution computed tomography (HRCT) pattern of RA-ILD, classified according to the standard criteria of the American Thoracic Society International Multidisciplinary Consensus Classification of the Idiopathic Interstitial Pneumonias [32], as follows: (1) UIP; (2) nonspecific interstitial pneumonia (NSIP); and (3) other (bronchiolitis, obliterans organizing pneumonia, lymphocytic interstitial pneumonitis, and mixed patterns). Progressive fibrosing ILD was defined according to the INBUILD criteria [33] and survival was assessed using data from Kaplan/Meier estimates, Cox Proportional Hazard models, Hazard Ratios (HRs), and Log-Rank Tests where applicable.

### 2.2. Search Strategy

Electronic databases, i.e., PubMed Medline and EMBASE, were searched using subject headings and text words related to the study population such as ‘interstitial lung disease’, ‘rheumatoid arthritis’, ‘prognosis’, ‘outcome’, and ‘mortality’. Methodology filters were not used to avoid limiting the sensitivity of the search. The Science Citation Index Expanded was also consulted using terms adapted from the previous search of Medline and EMBASE. Although no time limit was set, nearly all the articles were published from 1990 onwards. The references lists of the included studies and relevant review papers, and www.clinicaltrials.gov were also searched. Subsequently, a secondary manual search was carried out on the basis of the bibliographies of the initially selected articles. To address the risk of missing results, we evaluated study quality and examined the impact of incomplete data, although some biases were inevitable. To assess the included studies’ methodological quality, we used the Jadad quality scale for clinical trials [34] and the Oxford levels of evidence for observational studies [35]. The quality was then classified according to an asterisk system (* = low quality, ** = intermediate quality, and *** = high quality). The results were synthesized by the prognostic factor.

### 2.3. Data Extraction

Two reviewers (LG and CN) independently extracted relevant papers using a data extraction form that was modified from a previously published protocol paper for a systematic literature review (SLR) [22]. Any uncertainty or disagreement between the reviewers arising from these processes was resolved by discussion. The following data were extracted from each eligible study: the first author’s name, year of publication, study design, sample size and its demographic features, outcomes of interest, risk and prognostic factors, methods of statistical analysis, and summary statistics (Figure 1).

## 3. Results

### 3.1. Selection of Studies

Out of a total of 2217 records identified through a search of the two electronic databases, 85 records were retrieved as full-text after excluding 660 duplicates, 8 non-English reports, 1416 reports ineligible types, and 4 irrelevant articles, and finally, 48 studies were eligible for this review.

### 3.2. Demographic Features of Eligible Studies

Fifteen studies reported ILD progression, while the rest provided overall mortality data. The full results are available in Table A1. Among the selected studies, seven included subgroup analysis based on baseline disease activity [12,36,37,38], thirteen examined serological markers (rheumatoid factor-RF, anticitrullinated antibodies-ACPA, Kerbs von der Lungen-KL-6, interleukin-6-IL-6, and surfactant protein-D-SP-D) [9,38,39,40,41,42,43,44,45,46,47,48], eleven analyzed trajectories of pulmonary function tests [44,46,47,49,50,51,52,53,54,55,56], twenty-one studied the radiological pattern of ILD [9,10,16,39,43,44,48,50,52,53,57,58,59,60,61,62,63,64,65,66,67], and nine mentioned immunomodulatory treatments before ILD diagnosis [9,16,40,51,54,58,68,69,70]. A total of 1225 subjects with RA-ILD showed worsening lung function over time. Smoking history, reported in 37 studies, ranged from 5 to 85%, while the radiological and/or pathological UIP pattern, reported in 22 studies, ranged from 1.5 to 73%. The frequency of acute RA-ILD exacerbations ranged from 5.2% to 26.4% in three studies [58,71,72]. Subgroup analysis results should be interpreted with caution, as they were not the primary study focus, and many lacked *p*-values, making it difficult to draw firm conclusions about the observed effect.

### 3.3. Risk Factors of Lung Function Decline

Thirteen factors, including demographic variables, clinical risk factors (such as arthritis onset, clinical disease activity index—CDAI, and disease activity score—DAS28 scores), serum proteins (ACPA, KL-6, matrixmetaloproteinase 13-MMP13, and C-X-C motif chemokine 11-CSCL11/I-TAC), and preexisting treatments (corticosteroid and disease-modifying antirheumatic drugs—DMARDSs), were associated with RA-ILD progression. These factors were identified through univariate analyses (Table A2). Multivariate analyses most frequently identified age at initial presentation, RA disease activity, the percentage of the predicted diffuse capacity of the lungs for carbon monoxide—DLCO, and UIP pattern as significant risk factors for ILD prediction (Table 1).

### 3.4. Prognostic Factors for All-Cause Mortality of RA-ILD

Clinical data on all-cause mortality in RA-ILD were more prevalent than those on ILD progression, most of them reporting high mortality rates. Univariate analysis identified about 32 risk factors (Table A3), mainly related to RA disease activity (DAS28, CDAI, multidimensional health assessment questionnaire—MDHAQ, pain visual analoque scale-pain VAS, physician global assessment—PGA, and patient global assessment—PtGA), and HRCT pattern. In the multivariate analysis (Table 2), almost half of the prognostic factors remained significant. Age at initial presentation, male gender, disease duration, RA disease activity, elevated acute phase reactants, RF, PFTs, and a UIP pattern were significantly associated with all-cause mortality. The additional independent predictors of mortality were final oxygen saturation in the 6 min walking tests (6MWTs) [(HR 0.62, (95% CI 0.39–0.99)] [54], long-term exposure to particulate matter with an aerodynamic diameter of ≤10 µm (PM_10_) [HR 1.67 (95% CI 1.10–2.52)] [73], radiomics [HR 9.35 (95% 1.56–55.86)] [62], serum lactate dehydrogenase-LDH [HR 1.05 (95% CI 1.00–1.01)] [9], and specific serum biomarkers such as KL-6 [HR 3.23 (95% 1.39–7.51)], SP-D [HR 1.00 (95% CI 1.000–1.006)] [8,74], and IL-6 [HR 1.04 (95% 1.002–1.080)] [74].

### 3.5. Additional Analysis

Subgroup analysis was not undertaken due to the small number of the included studies. Sensitivity analysis could not be conducted because no studies were deemed as low risk of bias. Small study bias such as publication bias could not be assessed because the designated minimum number of studies was not available for any meta-analysis in this review.

## 4. Discussion

The current SLR underscores that ILD significantly impacts prognosis, contributing to increased morbidity and mortality in patients with RA. Clinical outcomes in RA-ILD patients showed considerable variation, with poor lung function outcomes observed in 1.5 to 85% of the cases and all-cause mortality ranging from 1.3 to 75%.

The multivariate analysis identified several key factors for ILD progression: age at RA diagnosis, male gender, smoking, high RF, serum KL-6, DAS28 score, lower baseline DLCO, and the presence of a UIP pattern on high-resolution computed tomography (HRCT). While RA-ILD often accompanies high RF and ACPA levels, only one study has demonstrated an association between ACPA and long-term function decline [39]. DLCO has been frequently associated with the acute exacerbations of lung disease and poorer outcomes compared to forced vital capacity—FVC. Several studies have indicated that DLCO serves as an early indicator of worsening lung function in RA-ILD patients [56,58,76]. However, there is notable heterogeneity among these studies, underscoring the need for further research into the relationship between PFTs and ILD progression.

Although five studies [16,40,51,57,68] linked corticosteroids and DMARDs to impaired lung function, these findings were based on small groups without comparators. Post hoc analyses suggest that certain poor prognostic factors may worsen ILD with methotrexate, leflunomide, or TNF inhibitors (TNFis). While some studies associated RA activity (DAS28) with ILD progression [36,67], a three-year study found no such link [77], hinting that immunosuppression could be beneficial, though more trials are needed. An analysis of 16 registries showed that seropositivity improves drug retention and response rates for non-TNFi biologic DMARDs, especially abatacept and rituximab, over tocilizumab [78]. Consequently, abatacept and rituximab are preferred as first-line biologics, with studies since 2018 suggesting they may stabilize or improve lung function in progressive ILD. Early research suggests JAK inhibitors could also be beneficial for RA-ILD, but further research is needed to confirm their safety and efficacy [62,79]. Additionally, Cano Jiménez et al. and Matson et al. found that discontinuing DMARDs negatively impacts survival, while immunosuppression improves lung function, even in patients with a UIP pattern on HRCT [54,80].

Equally important, numerous studies have identified significant predictors of mortality in RA-ILD patients: older age, male gender, reduced PFTs, and the presence of a UIP pattern on HRCT. Other factors linked to poorer survival include lower socioeconomic status, higher disease activity scores, and elevated ESR, although findings across studies have been variable [37,38,46,50]. The acute exacerbations of RA-ILD contribute directly to 10–20% of the mortality in RA, with a standardized mortality ratio of 2.5–5.0 compared to control populations [43]. A decline in the FVC of 10% consistently predicts death in retrospective RA-ILD cohorts and other non-IPF ILD groups, where radiologic progression of fibrosis also strongly indicates subsequent FVC decline [81]. In multivariable models, features like honeycombing [HR 2.49 (95% CI 1.09–5.69)] and combined pulmonary fibrosis and emphysema [HR 2.16 (95% 1.01–4.62)] independently predict mortality. A visual HRCT staging shows that ILD extent ≥20% increases mortality risk 3.78-fold in RA-ILD cohorts [57]. Similarly, in a Korean cohort of 153 RA-ILD patients, a visual scoring indicating ≥20% fibrosis is associated with a 4.5-fold risk of death in multivariable analysis [76]. Oh et al.’s quantitative lung fibrosis scoring on the HRCT images of 144 RA-ILD patients predicts worse 5-year mortality, with scores ≥12% of the total lung volume correlating with survival similar to IPF patients [50]. Radiomics, which quantifies computed tomography imaging features, was identified as a significant predictor for mortality [HR 9.35 (1.56–55.86)] but only in univariate analysis [62]. Regarding HRCT patterns, studies suggest that RA patients with a UIP pattern may experience poorer survival compared to those with NSIP or indeterminate patterns. Initial reports noted a 24% prevalence of UIP pattern in RA-ILD, associated with a disease trajectory similar to IPF and higher mortality rates [60]. However, in adjusted multivariable models considering baseline lung function and other confounders, the UIP pattern no longer independently correlates with increased mortality, suggesting complex relationships that warrant further investigation [44,51,82].

These findings carry significant implications for clinical practice, healthcare policy, and research. Clinicians should incorporate the identified factors into the assessment and management of RA-ILD patients, enabling more personalized treatment strategies. There is also a critical need for standardized guidelines to reduce variability in diagnosis and management. Research should prioritize the early detection of disease progression, particularly lung function decline, and focus on validating these prognostic factors.

The main limitation of this SLR is the inconsistency of the results, which limits confidence in the pooled estimates. Most studies were retrospective, with only 14 out of the 48 being prospective. Among these, six focused on the clinical course of RA-ILD, while the rest examined mortality predictors. The clinical diversity among the subjects, influenced by varying study objectives and criteria for RA-ILD diagnosis and classification, likely contributed to this inconsistency. Additionally, the inability to perform a meta-analysis due to significant heterogeneity in the study designs, patient populations, follow-up periods, and outcome definitions further limits our findings. This variability, along with the inconsistent reporting of prognostic factors, would have compromised the validity of a pooled analysis.

## 5. Conclusions

It is evident that ILD is a serious complication for patients with RA, and its mortality rate is significantly higher than that of patients with RA without ILD. Therefore, it is very important to know the prognostic factors of RA-ILD in advance for better treatment. This leads to a chance for early therapy and attentive follow-up, which could stop the progression of ILD and enhance the long-term result.

## Figures and Tables

**Figure 1 diagnostics-14-01890-f001:**
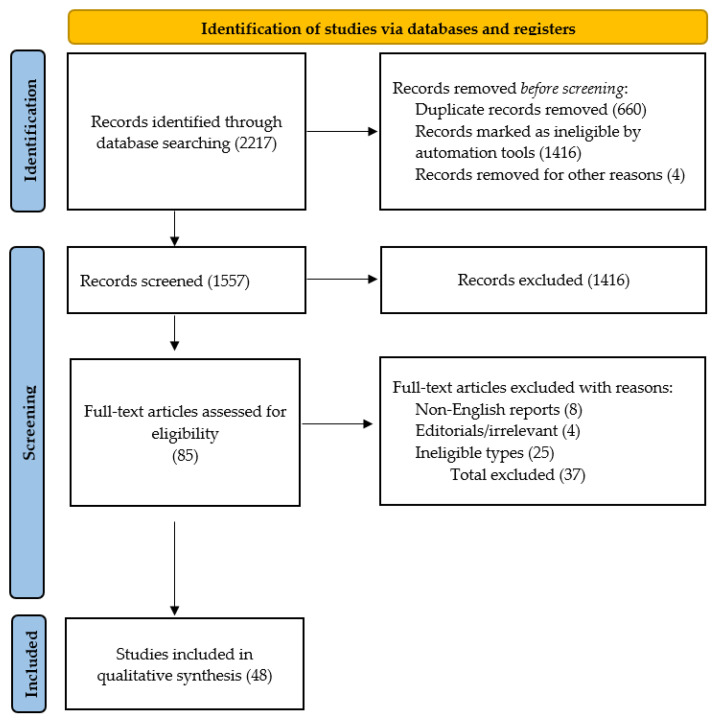
PRISMA flowchart: study selection process for RA-ILD prognosis and mortality review.

**Table 1 diagnostics-14-01890-t001:** Risk factors for ILD progression by multivariate analysis.

Potential Risk Factors	Study	Effect Estimate
Age at ILD diagnosis **	44, 70	HR 2.18; OR 1.7
Male gender **	70	OR 2.2
Smoking history ***	37, 70	OR 1.7–6.13
DAS28 ***	37	OR 1.71
Arthritis onset **	44	HR 1.87
KL-6 **	74	HR 3.37
DLCO **	53	OR 3.02
UIP pattern **	53	OR 3.47
Combined pulmonary fibrosis and emphysema **	59	OR 6.12
Preexisting rheumatic airway disease **	95	OR 7.40

DAS28: disease activity score 28; KL-6: Krebs von der Lungen; DLCO: diffusion capacity of the lung for carbon monoxide; UIP-usual interstitial pneumonia; **: moderate-quality evidence; ***: good-quality evidence.

**Table 2 diagnostics-14-01890-t002:** Prognostic factors for mortality in RA-ILD by multivariate analysis.

Prognostic Factor	References	Effect Estimate
Age **	[10,38,43,44,48,50,52,54,65,67]	HR 1.04–5.02
Male gender **	[70]	OR 2.5–18.13
Female gender **	[53]	HR 6.8
Smoking history *	[67]	HR 1.06–3.89
Disease duration of RA **	[50]	HR 1.3
ESR **	[44]	HR 5.35
HAQ disability **	[70]	OR 2.3
Steinbrocker class 3 or 4 **	[67]	HR 2.1
FVC% pred *	[44]	HR 2.52
DLCO ***	[49,54]	HR 0.85–0.97
TLCO **	[55]	HR 0.98
Final oxygen saturation in the 6MWT **	[54]	HR 0.62
UIP pattern **	[43,60,65,66,67]	HR 2.3–10.3
Non-UIP pattern **	[67]	HR 4.9
ILD extent ***	[52,53]	HR 2.40–9.01
Radiological honeycombing **	[48]	HR 3.69
Combined pulmonary fibrosis and emphysema **	[57]	HR 2.16
Pleural effusion **	[66]	HR 14.4
Corticosteroid *	[70]	HR 2.5
Immunosuppressive agents **	[70]	HR 3.0
Withdrawal of MTX or LFN after ILD diagnosis **	[54]	HR 2.18
Diagnostic delay of ILD **	[54]	HR 1.11
PM_10_ *	[73]	HR 1.67
History of acute ILD exacerbations ***	[48,75]	HR 2.42–6.48

RA—rheumatoid arthritis; ESR—erythrocyte sedimentation rate; HAQ—health assessment questionnaire; FVC—forced vital capacity; TLCO—transfer factor for carbon monoxide; DLCO—diffusion capacity of the lung for carbon monoxide; 6MWT—6 min walking test; UIP—usual interstitial pneumonia; ILD—interstitial lung disease; MTX—methotrexate; LFL—leflunomide; PM_10_—particulate matter with an aerodynamic diameter of ≤10 µm; HR: Hazard Ratio; OR: Odds Ratio; CI: Confidence Interval; *: low-quality evidence; **: moderate-quality evidence; ***: good-quality evidence.

## Data Availability

No new data were created or analyzed in this study. Data sharing is not applicable to this article.

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
