# Peer review of "A Systematic Review of the Key Predictors of Progression and Mortality of Rheumatoid Arthritis-Associated Interstitial Lung Disease"

_diagnostics, 2024, doi:10.3390/diagnostics14171890_

Round 1

Reviewer 1 Report

Comments and Suggestions for Authors

Comments on the Quality of English Language

The English can be improved (see comments)

Author Response

Thank you for your email and for providing the detailed checklist. I am pleased to inform you that I have revised the manuscript in accordance with the reviewers' comments.

  1. I have ensured that all references are relevant to the content of the manuscript.
  2. All revisions have been highlighted to allow editors and reviewers to easily identify the changes made.
  3. A cover letter has been prepared to respond to each of the reviewers’ comments and to explain the manuscript revisions point by point.
  4. I have carefully considered the recommended references, including only those that enhance the manuscript.
  5. For any comments that were not fully addressed, I have included a detailed explanation in my response.

The revised manuscript and the cover letter have been uploaded for your review.

Please let me know if there are any further steps required.

Thank you for your time and assistance.

Reviewer 2 Report

Comments and Suggestions for Authors

The choice of the topic by the authors is very appropriate, because it is currently one of the unsolved issues in RA.

Author Response

Dear Reviewer,

Thank you for your kind words and positive feedback on our manuscript. We are pleased that you find the topic relevant, especially given its significance as one of the unsolved issues in RA.

We appreciate your recognition of our work, and we are committed to further refining the manuscript to ensure it contributes valuable insights to the field.

Reviewer 3 Report

Comments and Suggestions for Authors

The authors comprehensively analyzed the major factors influencing the course and death from interstitial lung disease linked to rheumatoid arthritis.

It is well-written; however, it may need a few clarifications before being considered for publication.

General comments

- Spell out the full term at its first mention, indicate its abbreviation in parenthesis, and use the abbreviation from then on.

Introduction

- You may highlight the recent epidemiology of RA, including its prevalence and female-to-male ratio. DOI: 10.1177/03000605231204477

- You may highlight the prevalence of RA-ILD varies according to the detection methods used and cohort studied. DOI: 10.1038/s41598-020-72768-z

 Materials and Methods

- Describe any methods used to assess the risk of bias due to missing results in a synthesis

Results

In Table 1, you describe the observational perspective of the Mena-Vázquez 2020 study, which was done from 2015 to 2017; however, you cited another study by the same author, which was done from 2015 to 2023 and was published in 2024.

Please make the necessary corrections.

 Discussion

- Discuss the implications of the results for practice, policy, and future research.

- We should fully discuss the implications of the results for using bDMARDs, e.g., TNF-alpha inhibitors, IL-6 inhibitors, Rituximab, cDMARDs, e.g., methotrexate, and leflunomide in RA-ILD patients.

Conclusion

- The takeaway message seems weak. How can the readers apply these findings to their daily practice?

Author Response

Dear Reviewer,

Thank you for your thorough and constructive feedback on our manuscript. We greatly appreciate your positive remarks and have carefully addressed all the issues you raised.

  • We have ensured that all terms are spelled out in full at their first mention, with abbreviations provided in parentheses and used consistently throughout the manuscript.
  • The Introduction now includes a discussion of the recent epidemiology of RA, highlighting its prevalence and the female-to-male ratio, as well as the variability in RA-ILD prevalence based on detection methods and cohort studies, as per your recommendations.
  • In the Materials and Methods section, we have added a description of the methods used to assess the risk of bias due to missing results in the synthesis.
  • We corrected the reference to the Mena-Vázquez study in Table 1 to accurately reflect the study period and publication details.
  • The Discussion has been expanded to cover the implications of our results for clinical practice, policy, and future research. We have also thoroughly discussed the use of bDMARDs, such as TNF-alpha inhibitors, IL-6 inhibitors, Rituximab, and cDMARDs like methotrexate and leflunomide in RA-ILD patients.
  • Finally, we have strengthened the Conclusion to offer a clearer takeaway message, providing guidance on how readers can apply our findings in daily practice.

We believe these revisions have improved the manuscript, and we hope it now meets your expectations for publication. Thank you once again for your valuable feedback.

Round 2

Reviewer 3 Report

Comments and Suggestions for Authors

The article has been substantially improved. Now it is ready to be published.